# One Advantage of Being Polyploid: Prokaryotes of Various Phylogenetic Groups Can Grow in the Absence of an Environmental Phosphate Source at the Expense of Their High Genome Copy Numbers

**DOI:** 10.3390/microorganisms11092267

**Published:** 2023-09-09

**Authors:** Patrik Brück, Daniel Wasser, Jörg Soppa

**Affiliations:** Biocentre, Institute for Molecular Biosciences, Goethe University, Max-von-Laue-Str. 9, D-60438 Frankfurt, Germanywasser@bio.uni-frankfurt.de (D.W.)

**Keywords:** polyploidy, phosphate starvation, phosphate storage polymer, *Zymomonas mobilis*, *Azotobacter vinelandii*, *Halobacterium salinarum*, *Synechocystis* PCC 6803, *Haloferax volcanii*, *Escherichia coli*

## Abstract

Genomic DNA has high phosphate content; therefore, monoploid prokaryotes need an external phosphate source or an internal phosphate storage polymer for replication and cell division. For two polyploid prokaryotic species, the halophilic archaeon *Haloferax volcanii* and the cyanobacterium *Synechocystis* PCC 6803, it has been reported that they can grow in the absence of an external phosphate source by reducing the genome copy number per cell. To unravel whether this feature might be widespread in and typical for polyploid prokaryotes, three additional polyploid prokaryotic species were analyzed in the present study, i.e., the alphaproteobacterium *Zymomonas mobilis*, the gammaproteobacterium *Azotobacter vinelandii*, and the haloarchaeon *Halobacterium salinarum.* Polyploid cultures were incubated in the presence and in the absence of external phosphate, growth was recorded, and genome copy numbers per cell were quantified. Limited growth in the absence of phosphate was observed for all three species. Phosphate was added to phosphate-starved cultures to verify that the cells were still viable and growth-competent. Remarkably, stationary-phase cells grown in the absence or presence of phosphate did not become monoploid but stayed oligoploid with about five genome copies per cell. As a negative control, it was shown that monoploid *Escherichia coli* cultures did not exhibit any growth in the absence of phosphate. Taken together, all five polyploid prokaryotic species that have been characterized until now can grow in the absence of environmental phosphate by reducing their genome copy numbers, indicating that cell proliferation outperforms other evolutionary advantages of polyploidy.

## 1. Introduction

For many years, it has been believed that archaea and bacteria (prokaryotes) are typically monoploid (often called haploid) and contain one copy of a circular chromosome [1]. It was known that some species like *Deinococcus radiodurans* contain multiple copies of their chromosome [2], but these species were thought to be a rare exception to the rule. However, over the last two decades, it has been reported that many prokaryotic species of various phylogenetic groups are not monoploid, but oligoploid (up to ten genome copies), polyploid (up to 100 genome copies), or hyperpolyploid (more than 100 genome copies). This was shown to be true for representatives of the proteobacteria [3], cyanobacteria [4,5], Firmicutes [6], *Thermus* [7], methanogenic archaea [8], and halophilic archaea [9,10]. In addition, most or all giant bacteria of different phylogenetic groups seem to be polyploid [11,12]. Examples are *Epulopiscium*, which contains tens of thousands of genome copies [13], and *Achromatium oxaliferum*, the only known naturally occurring heterozygous bacterium [14]. Taken together, currently, it seems that the majority of prokaryotic species contain more than one copy of the genome and thus are not monoploid.

Various evolutionary advantages of polyploidy for prokaryotes have been proposed, several of which have been addressed experimentally [15,16]. For giant bacteria, it has been proposed that their large cell size is incompatible with monoploidy, because diffusion limits would inhibit the distribution of transcripts and proteins throughout the cell volume if only a single chromosome would be present. The most obvious advantage for normal-sized cells are a low mutation rate and high resistance against double-strand breaks (DSBs), because damaged copies can be repaired using wild-type templates. This has been most intensely studied with *D. radiodurans*, which has very high resistance against X-ray radiation and desiccation (also inducing DSBs). *D. radiodurans* can restore intact chromosomes from hundreds of chromosomal fragments after chromosomes have been scattered by exposure to one of the two conditions inducing DSBs [17,18,19]. High radiation resistance has also been reported for *Halobacterium salinarum* [20]. In fact, it has been reported that the ability to repair DSBs in haloarchaea is so high that they can survive millions of years in salt deposits [21,22]. It has been shown that in many species, the number of chromosomes is highly regulated in response to growth phase, growth rate, and/or physical and chemical conditions of the environment [9,23]. For many species, the chromosome copy number is higher in exponential phase than in stationary phase [3,6,8,15], and this gene dosage regulation has been proposed to be a mechanism of global regulation of gene expression. However, the opposite regulation has been also reported, e.g., the methanogenic archaeon *Methanosarcina acetivorans* during slow growth and the proteobacterium *Azotobacter vinelandii* have been described as containing more chromosomes in stationary than in exponential phase [8,24], i.e., 2.5 versus 5 copies for *M. acetivorans* and 4 versus 80 copies for *A. vinelandii*. For the cyanobacterium *Synechocystis* PCC 6803, the number of genome copies was influenced by the light intensity as well as by the external phosphate concentration [23], e.g., chromosome copy number doubled from 27.0 to 53.4 with low light intensity. In summary, the distribution of ploidy levels indicates that polyploidy has evolved in different phylogenetic groups independently at different times in evolution and been driven by different evolutionary advantages.

An additional putative evolutionary advantage of polyploid prokaryotes over monoploid species is the ability to grow in the absence of an external phosphate source. Many environments are phosphate-limited and exhibit large fluctuations in phosphate concentrations [25]. Species that could multiply in the absence of phosphate would outnumber species without this ability during times of phosphate starvation and could outgrow them when phosphate becomes available again. And indeed, for the cyanobacterium *Synechosystic* PCC 6803 and the halophilic archaeon *H. volcanii*, it has been reported that they can grow in the absence of an external phosphate source while the genome copy number per cell decreases [9,23]. For *H. volcanii*, the decrease in chromosome copy number was larger than the increase in cell number; therefore, it has been proposed that genomes might be degraded during phosphate starvation to liberate phosphate for other phosphate-containing cellular molecules, and genomic DNA might be a bona fide phosphate storage polymer [9]. Supporting this view, readdition of phosphate to a phosphate-starved culture led to a very rapid increase in the genome copy number with a transient overshoot to 40 genomes per cell, a number twice as high as during exponential steady-state growth.

The present study aimed to elucidate whether additional polyploid prokaryotes also have the ability to grow in the absence of external phosphate and whether this feature might therefore be a more widespread and typical evolutionary advantage of polyploid prokaryotes. To this end, three species were chosen, the alphaproteobacterium *Zymomonas mobilis,* the gammaproteobacterium *A. vinelandii* (the same group as *E. coli*), and the halophilic archaeon *H. salinarum*. As a negative control, slowly growing monoploid *E. coli* cultures were also analyzed.

The first species was the alphaproteobacterium *Z. mobilis*. *Z. mobilis* was isolated more than a century ago and was known under the names *Termobacterium mobilis* and *Pseudomonas mobilis* before its current name was adopted [26]. In contrast to the obligate aerobe *A. vinelandii* (see below), *Z. mobilis* grows anaerobically; therefore, aerobic as well as anaerobic lifestyles were present in our study. *Z. mobilis* is of specific interest because of its high potential for biotechnological applications. It can convert simple sugars into ethanol at nearly the maximal theoretical yield, with a higher efficiency than *Sacchomyces cerevisiae* [27,28]. However, in addition to *Z. mobilis* being applied to produce ethanol, its metabolism is also designed for the production of a wide range of different products [29,30,31,32]. A recent study revealed that *Z. mobilis* is polyploid with about 20 genome copies throughout growth [33].

The second species, *Azotobacter vinelandii,* was isolated 120 years ago from a soil in Vineland, New Jersey, USA [34]. It is characterized by an obligate aerobic lifestyle, the ability to fix molecular nitrogen, natural competence, production of polyhydroxyalkanoates and alginates, and formation of desiccation-resistant cysts that can survive for decades in dry soil [34]. It has a very versatile metabolism and can grow on a variety of different carbon sources and at a wide range of oxygen concentrations. In particular, aerobic nitrogen fixation and its regulation has been studied very intensively because the *A. vinelandiii* nitrogenases, like all nitrogenases, are oxygen sensitive [35,36]. Nitrogen fixation as well as additional features make *A. vinelandii* a plant growth-promoting bacterium of the rhizosphere [34]. In addition, biotechnological applications have been proposed, e.g., the production of biodegradable polymers and the usage of alginate as wound-dressing material in medicine [34]. Concerning the ploidy level, *A. vinelandii* is a special case because contradicting reports have been published. Using flow cytometry, it was revealed that in complex medium, early exponential phase cells were oligoploid with about four copies of chromosomes, and that the chromosome number steadily increases during growth, with about 40 copies of the chromosome in late-exponential cells and 80 copies in early stationary-phase cells [24]. In stark contrast, several genetic analyses indicated that *A. vinelandii* is not polyploid, but, e.g., segregation patterns of mutations indicated that it is a monoploid bacterium [37,38,39]. This discrepancy prompted us to include *A. vinelandii* in the present study to clarify its chromosome copy number, and—if polyploid—to elucidate whether it can grow in the absence of external phosphate.

The third species was the halophilic archaeon *H. salinarum*. *H. salinarum* was isolated 100 years ago from salted fish [40]. This isolate was subsequently lost, and the type strains that are used today were isolated in 1934 from salted animal hides [41]. *H. salinarum* contains the four retinal proteins bacteriorhodopsin, halorhodopsin, and sensory rhodopsins I and II, which have been intensively studied [42,43,44,45]. Bacteriorhodopsin is a light-driven proton pump and mediates the second principle mechanism of photosynthesis, which is very different from the electron transport-dependent photosynthesis of bacteria and plants. Halorhodopsin is a light-driven chloride pump that is important for osmoadaptation, and the two sensory rhodopsins are involved in positive and negative phototaxis. *H. salinarum* also harbors gas vesicles, which enable the cells to float at the top of salt ponds, where bacteriorhodopsin-driven photosynthesis is efficient and the oxygen concentration is higher than in the water column [46]. *H. salinarum* was found to be polyploid, with more than 20 copies of the major chromosome throughout the growth curve during aerobic as well as anaerobic growth [10].

The fourth species was *Escherichia coli*, by far the most studied model bacterium, with more than 300,000 publications with “*Escherichia coli*” in the title or abstract in PubMed. *E. coli* can grow very fast with a doubling time of around 20 min under optimal conditions. This is much shorter than the time needed for replication and segregation of the chromosome. Therefore, *E. coli* switches to multifork replication under these conditions and contains on average 6.8 origins of replication and 1.7 termini, which is called mero-oligoploid [3]. However, during slow growth in synthetic medium with a generation time of about two hours, *E. coli* becomes monoploid. Therefore, slow-growing *E. coli* cultures were used in this study as a negative control.

The experimental design of this study was to grow the three polyploid species in complex medium, remove the medium, and then incubate the cells in synthetic medium in the presence and in the absence of external phosphate. After growth of the phosphate-starved cells had ceased, phosphate was added to verify that the cells were still alive and to elucidate whether an overshoot of the genome copy number occurred. Throughout the experiment, growth was followed by counting the cells using a Neubauer counting chamber, and the average genome copy number per cell was quantified using qPCR.

## 2. Materials and Methods

### 2.1. The qPCR Method for the Quantification of Chromosome Copy Numbers

The chromosome copy numbers were quantified using the real-time PCR (qPCR) method that was established about 15 years ago [10] and that has been applied to many species of different phylogenetic groups [4,6,8,21,23,33,47]. During the first studies, the qPCR method was validated against two independent, alternative methods, i.e., quantitative Southern [10] blotting and spectroscopic DNA quantification [8]. Appendix A provides an overview of the steps of the method. The following steps must be optimized for every new species under investigation: (1) The cell disruption method has to guarantee that at least about 90% of all cells are lysed, while the genomic DNA remains mainly intact. The optimal cell disruption method varies from species to species. (2) Several primers have to be tested for each genomic site to find the optimal combination that guarantees exponential amplification of a single PCR product. (3) It has to be verified that the cytoplasmic extract does not inhibit the qPCR but enables true exponential amplification (if necessary, it has to be dialyzed prior to its addition to the qPCR). This point can be verified by comparing the differences in C_T_ values of serial dilutions; a difference of 3.32 for a tenfold dilution shows that the PCR is truly exponential (2^3.32^ = 10), and the results can be used to quantify the chromosome copy numbers. This optimization was performed for every species used in this study. The optimized parameters for every species are described in following paragraphs. The primer pairs that were used are summarized in Appendix A. Each experiment was performed in three biological replicates, and average values and their standard deviation are reported.

### 2.2. Cultivation and Cell Disruption of Zymomonas mobilis

*Z. mobilis* strain Zm4 was grown anaerobically in complex and in synthetic medium as described previously [28,33]. Quantities of 12 mL medium in 15 mL screw-cap tubes were inoculated, flushed with N_2_, and incubated at 30 °C with shaking (200 rpm) to inhibit cell sedimentation.

Complex medium contained per liter: 20 g glucose, 5 g yeast extract, 1 g ammonium sulfate, 1 g KH_2_PO_4_, 1 g MgSO_4_ × 7 H_2_O.

Synthetic medium contained per liter: 20 g glucose, 10 g MgSO_4_ × 7 H_2_O, 0.5 g NaCl, 1 g (NH_2_)_2_SO_4_, 0.2 g CaCl_2_, 10 mM MES pH 5.9, 1 mL BME vitamin solution, 25 mg Na_2_MoO_4_. MES was used as a buffer instead of phosphate to enable phosphate starvation experiments.

The starvation experiments were performed in 24 mL medium in 30 mL screw-cap tubes. The control cultures contained 14 mM phosphate (50 × stock solution: 350 mM K_2_HPO_4_, 350 mM KH_2_PO_4_), the same concentration was added to the starved culture at the indicated time.

At the desired time points, cells were harvested using centrifugation and resuspended in 1 mL TE solution (10 mM Tris/HCl pH 8, 2.5 mM EDTA). The suspensions were transferred to 2 mL tubes that contained 0.5 g silica beads (0.1 mm, Roth, Karlsruhe, Germany). Cells were disrupted with a Speed Mill (Analytik Jena, Jena, Germany) with 7 cycles of 30 s shaking in a cold room. Cell debris was removed using centrifugation, and the supernatant (cell extract) was used for qPCR without further treatment. Microscopic analysis prior to and after the procedure revealed that 85.5 ± 4.1% of all cells were disrupted.

### 2.3. Cultivation and Cell Disruption of Azotobacter vinelandii

*A. vinelandii* strain DSM 2289 was grown in complex and in synthetic medium as described previously described [48]. Quantities of 25 mL cultures were grown in 100 mL Erlenmeyer flasks at 30 °C and 200 rpm.

Per liter, the complex medium contained 1 g tryptone, 0.5 g yeast extract, 0.5 g ammonium acetate, 6.7 mM phosphate (100 × stock solution: 500 mM K_2_HPO_4_, 160 mM KH_2_PO_4_), 300 µM CaCl_2_, 400 µM MgSO_4_, 8 µM FeSO_4_, 100 µg Na_2_MoO_4_. In addition, either 0.5 g glucose or 0.25 g glucose/0.25 g mannitol were added.

Per liter, Burk’s synthetic medium for *A. vinelandii* contained either 0.5 g glucose or 0.25 g glucose/0.25 g mannitol, 20 mM Tris/HCl pH 7.2, 300 µM µM CaCl_2_, 400 µM MgSO_4_, 8 µM FeSO_4_, 100 µg Na_2_MoO_4_.

For the analysis of the ideal phosphate concentration, different concentrations were added as indicated. For the phosphate starvation experiment 1 mM phosphate was added to the control culture at the beginning, and to the phosphate-starved culture at the indicated time point.

At the desired time points cells were harvested using centrifugation and resuspended in 1 mL TE solution (10 mM Tris/HCl pH 8, 2.5 mM EDTA). The suspensions were transferred to 2 mL tubes that contained 0.5 g silica beads (0.1 mm, Roth, Karlsruhe, Germany). Cells were disrupted with a Speed Mill (Analytik Jena, Jena, Germany) with 7 cycles of 30 s shaking in a cold room. Cell debris was removed using centrifugation, and the supernatant (cell extract) was used for qPCR without further treatment. Microscopic analysis prior to and after the procedure revealed that 88.2 ± 3.9% of all cells were disrupted.

### 2.4. Cultivation and Cell Disruption of Halobacterium salinarum

*H. salinarum* strain DSM 670 was grown in complex and synthetic medium as described previously [10,49,50]. Quantities of 30 mL cultures were grown in 100 mL Erlenmeyer flasks at 42 °C with shaking (200 rpm).

Per liter, the complex medium contained 10 g peptone, 250 g NaCl, 3 g tri-sodium citrate dehydrate, 2 g KCl, 20 g MgSO_4_ × 7 H_2_O, pH 7.0–7.2.

Per liter, the synthetic medium contained 900 mL saline, 50 mL amino acid stock solution, 1 mL trace elements solution, 1 mL BME vitamin solution, 20 mM Tris/HCl pH 7.2. The 900 mL saline solution contained 234 g NaCl, 0.1 g KNO_3_, 2 g KCl, 20 g MgSO_4_ × 7 H_2_O, 0.5 g tri-sodium citrate dehydrate. Quantities of 100 mL amino acid stock solution contained 0.8 g aspartate, 3.97 g glutamate, 1.6 g leucine, 0.15 g glycine, 0.45 g alanine, 1.22 g arginine, 0.89 g isoleucine, 0.39 g methionine, 0.21 g proline, 0.26 g phenylalanine, 1.22 g serine, 1.0 g threonine, 0.4 g tyrosine, 0.59 g valine, 0.61 g lysine. Per 50 mL the 1000× trace element stock solution contained 161.3 mg FeSO_4_ × 7 H_2_O, 2.5 mg CuSO_4_ × 5 H_2_O, 17.8 mg MnCl_2_, 21.6 mg ZnSO_4_, 1.2 mg Na_2_MoO_4_ × 2 H_2_O. When desired, 1 mL of a phosphate stock solution was added (0.42 M K_2_HPO_4_, 0.58 M KH_2_PO_4_).

For the quantification of chromosome copy numbers, 2 mL of culture were centrifuged, and the pellet was resuspended in 1 mL of basal salts (medium without carbon source). Quantities of 200 µL cell suspension were added to 1.8 mL of distilled water, which resulted in rapid cell lysis due to the osmotic shock.

### 2.5. Cultivation and Cell Disruption of Escherichia coli

*E. coli* strain MG1655 was grown in M9 synthetic medium as described previously [3,51]. Quantities of 25 mL cultures were grown in 100 mL Erlenmeyer flasks at 37 °C with shaking (200 rpm).

Per liter, the M9 medium contained 4 g sodium succinate, 1 mM MgSO_4_, 40 µM CaCl_2_, 20 mM Tris/HCl pH 7.2, and 200 mL 5 × M9 salt solution. Tris was used as a buffer instead of the original phosphate buffer to allow phosphate starvation experiments. Per liter, the 5 × M9 salt solution contained 2.5 g NaCl, and 5 g NH_4_Cl. For phosphate-containing media 10 mL of phosphate stock solution was added per 1 l medium (phosphate stock solution: 686 mM Na_2_HPO_4_, 314 mM KH_2_PO_4_).

At the desired time points cells were harvested using centrifugation and resuspended in 1 mL TE solution (10 mM Tris/HCl pH 8, 2.5 mM EDTA). The suspensions were transferred to 2 mL tubes that contained 0.5 g silica beads (0.1 mm, Roth, Karlsruhe, Germany). Cells were disrupted with a Speed Mill (Analytik Jena, Jena, Germany) with 8 cycles of 30 s shaking in a cold room. Cell debris was removed using centrifugation, and the supernatant (cell extract) was used for qPCR without further treatment. Microscopic analysis prior to and after the procedure revealed that 92.0% ± 3.0% of all cells were disrupted.

### 2.6. Growth Measurements

To monitor growth, aliquots of the cultures were removed, and the cell densities were quantified using a Neubauer counting chamber. The cell density was calculated using the following formula: cell density (cells/mL) = No. cells per small square × 2 × 10^7^. At least 50 cells were counted to reach statistical significance. Cell counting was used to monitor growth instead of OD measurements to enable the calculation of chromosome copy numbers per cell. The known cell densities of the samples enabled the calculation of how many cells were used to quantify the number of chromosomes (see below), and the division of the number of chromosomes by the number of cells yielded the number of chromosomes per cell.

### 2.7. Quantification of Average Genome Copy Numbers

For each species, genomic DNA was isolated using standard procedures, which was used as template in a PCR reaction to generate a standard fragment of about 1 kbp. The oligonucleotides and the applied annealing temperatures are listed in Appendix A. Analytical agarose gels were used to verify that a single PCR fragment of the desired size had been amplified. The standard fragments were purified using the GenElute PCR-Clean-Up kit (Sigma-Aldrich, St. Louis, MO, USA). The concentrations of the standard fragments were quantified using a Nanodrop spectrophotometer. The number of DNA molecules per unit volume were calculated using the “SMS DNA Molceular Weight tool” (www.bioinformatics.org/sms2 (accessed on 30 June 2023)).

The cytoplasmic extracts were generated as described above. For each species, serial dilutions of the standard fragment and the cytoplasmic extract were analyzed simultaneously in a real-time PCR (qPCR) reaction using the RotorGene 3000. For the standard fragment, dilutions from 10^−3^ to 10^−8^ were used; for the cytoplasmic extract, 10-fold and 100-fold dilutions were used. Three technical replicates were analyzed for each dilution. A “no template control” was included as a negative control. The amplified “analysis fragments” had sizes of about 200 nt–300 nt. The C_T_ values (threshold values) of the dilution series of the standard fragment were used to generate a standard curve, which was used to quantify the molecules per unit volume from the C_T_ values of the cytoplasmic extracts. Together with the cell numbers used for extract generation, these values were used to calculate the copy numbers per cell. Average values of three biological replicates, each with at least six technical replicates, were used to calculate average values and their standard deviations.

### 2.8. Data Bases and Bioinformatic Analyses

The genomes of the four species were retrieved from the National Library of Medicine (www.ncbi.nih.gov (accessed on 23 January 2023)). The positions of replication origins were determined using the DoriC database version 12.0 (www.tubic.org/doric (accessed on 23 January 2023)). The software “clone manager professional suite” version 8 (Scientific and Educational software, Denver, CO, USA) was used for planning the experimental designs, including primer sequences. The theoretical melting points of primers were calculated using the “NEB Tm calculator” (www.tmcalculator.neb.com (accessed on 31 July 2023)). The results were calculated, and the figures were prepared using GraphPad Prism version 5 (GraphPad software, San Diego, CA, USA).

## 3. Results

### 3.1. The qPCR Method for Genome Copy Number Quantification

The number of genome copies per cell were quantified making use of the real-time PCR (qPCR) method, which was established about 15 years ago for halophilic archaea [10] and has been applied for the characterization of various species since then [3,4,6,8,23,33,47,52]. In short, a standard fragment of about 1 kbp is generated for the genomic site that should be quantified. Then, the species is grown to the desired metabolic state (e.g., exponential phase, stationary phase) and a cytoplasmic extract is generated from a defined number of cells. Dilution series of the standard fragment and the cytoplasmic extract are analyzed simultaneously using qPCR. The difference in C_T_ values between tenfold dilutions must be around 3.32 to guarantee that all reactions were truly exponential and allow a quantitative analysis. A standard curve is generated and used to quantify the copy number of the genomic site under investigation, which is used to calculate the copy number per cell. An overview of the method is provided in Appendix A. Several steps of the method were optimized for each of the four species that were analyzed in the current study (see Methods), i.e., the polypoid bacteria *Z. mobilis* and *A. vinelandii*, the polyploid haloarchaeon *H. salinarum*, and the monoploid bacterium *E. coli* used as a negative control.

### 3.2. Characterization of Growth and Genome Copy Numbers of Zymomonas mobilis

For the phosphate starvation experiment a preculture of *Z. mobilis* Zm4 was grown anaerobically in complex medium. It was washed free of medium, and two aliquots were used to inoculate synthetic medium with 2% (*w*/*v*) glucose as sole carbon and energy source. The control culture contained 14 mM phosphate, while the test culture lacked any phosphate source. The control culture was grown for 30 h through exponential and transition phase into stationary phase. The test culture was incubated for 24 h in the absence of phosphate until growth had ceased. Then, 14 mM phosphate was added, and the culture was incubated for another 50 h until it reached stationary phase. Three biological replicates were performed, and average results and their standard deviations are shown in Figure 1A (control culture) and Figure 1B (test culture). At the indicated five time points, the chromosome copy number was quantified. The preculture had an average copy number of 15. The copy number was downregulated in the control culture to about five during exponential growth and in stationary phase. The doubling time was 3.9 h, nearly twice as high as the doubling time of 2 h of the preculture grown in complex medium. These results show that the chromosome copy number in *Z. mobilis* is higher in faster than in slower growing cells.

Remarkably, the cell number of the test culture nearly tripled in the absence of phosphate, showing that limited growth is possible without any external phosphate source. At the same time, the average chromosome number per cell decreased by a factor of three, indicating that replication was impossible in the absence of an external phosphate source, but that the preexisting chromosomes were segregated to the daughter cells. After the addition of 14 mM phosphate, the culture started rapid exponential growth after a lag phase of 20 h. As it had reached stationary phase, it ended up with the identical cell density and the same ploidy level of five copies per cell as the control culture.

### 3.3. Characterization of Growth and Genome Copy Numbers of Azotobacter vinelandii

Prior to the phosphate starvation experiment, the genome copy number of *A. vinelandii* in different media was clarified. First, the cells were grown in Burk’s complex medium supplemented with glucose and mannitol [48]. The cells had average chromosome copy numbers of 20.6 during exponential phase and 19.7 during early stationary phase. Thus, it was confirmed that *A. vinelandii* is polyploid during growth in complex medium, albeit we did not find the reported very high copy numbers of up to 80 [24].

Next, this preculture was used to inoculate Burk’s synthetic medium with glucose and mannitol (0.25% (*w*/*v*) each) as sole carbon and energy source and four different phosphate concentrations, from 0.1 mM to 10 mM (Figure 2). Concerning the genome copy numbers, the results were very similar for all the four phosphate concentrations, which span a range of 100-fold: at the end of lag-phase/onset of growth, the copy number considerably increased from the 20 copies present in the inoculum to about 40 copies. Then, the copy number sharply dropped to 10 copies already during exponential growth, and it dropped to less than 10 in stationary phase. The only difference was found in the growth yield, while the culture reached a cell density of 5 × 10^8^ cells/mL in medium with 0.1 mM phosphate, the cell density in stationary phase was 1.5 × 10^9^ cells/mL in medium with the other three phosphate concentrations. It is remarkable that the 100-fold difference in external phosphate concentration hardly influenced the internal genome copy number, while the copy number was differentially regulated about fivefold during the growth curve. Based on these results, it was decided to use 1 mM phosphate for the phosphate starvation experiment.

The results of the control culture have already been described above, and they are shown again in a different form in Figure 3A to enable a good comparison with the test culture. The test culture was incubated for 40 h in the absence of external phosphate (Figure 3B). During that time, the average cell density increased by a factor of 5.4, while the average chromosome copy number decreased by a factor of 4.9 After the addition of 1 mM phosphate, the test culture started to grow exponentially, and it reached the same cell density of 1.5 × 10^9^ cells/mL and the same chromosome copy number of about five in stationary phase as the control culture (Figure 3B).

The phosphate starvation experiment with *A. vinelandii* was also performed with sucrose as an alternative carbon source. The preculture grown in Burk’s complex medium with added sucrose (0.5% (*w*/*v*)) had an average copy number of 31.4, considerably higher than the 20.6 copies in the complex medium with added glucose and mannitol (see above). The control culture in Burk’s synthetic medium with sucrose and 1 mM phosphate reached a final density of 2 × 10^9^ cells/mL in stationary phase, slightly higher than the control culture with glucose/mannitol (Figure 4A). Concomitantly, the chromosome copy number dropped to 3.5, slightly lower than with glucose/mannitol. The test culture was incubated for 40 h in the absence of external phosphate (Figure 4B). During this time, the cell density increased by a factor of 6.7, and the chromosome copy number dropped by a factor of 7.4 down to 4.2 copies per cell. After the addition of 1 mM phosphate to the starved culture, exponential growth started, and the culture reached the same cell density and the same chromosome copy number in stationary phase as the control culture. Together, the results show that the carbon source has a slight effect on the growth yield and the chromosome copy number of *A. vinelandii*, but that qualitatively the results are very similar. Similar to *Z. mobilis* (see above), the cells did not become monoploid, neither during phosphate starvation, nor in stationary phase after exponential growth in the presence of phosphate.

### 3.4. Characterization of Growth and Genome Copy Numbers of Halobacterium salinarum

For *H. salinarum*, it has been reported that it is polyploid, and that the copy number of its major chromosome is growth phase-regulated, with a higher copy number during mid-exponential growth phase and a copy number decrease in stationary phase [10]. However, only a single site was characterized, but the major chromosome of *H. salinarum* contains three origins of replication, in contrast to the two polyploid bacteria described above. Therefore, the experiment was repeated with the quantification of the copy numbers of four sites distributed around the chromosome, i.e., one site near to oriC1 (defined as 0% of the chromosome), a second site at 33% around the chromosome, a third site at 66% of the chromosome, and a fourth site near oriC3, at about 90% of the chromosome (Appendix A). Thus, two sites were close to one of the three origins or replication, while two sites were far from an origin. Three biological replicates were grown in complex medium, and cell densities and copy numbers of the three genomic sites were determined. As shown in Figure 5, the copy numbers of the four sites were identical throughout the growth curve. The copy numbers were upregulated during exponential growth and downregulated in stationary phase, as was reported before for the site near oriC3. Based on these results, we decided that it was enough to quantify one site during the phosphate starvation experiment, and the site near oriC3 was chosen.

For the phosphate starvation experiment, precultures of *H. salinarum* were grown in complex medium, washed free of medium, and aliquots were used to inoculate three biological replicates in synthetic medium with 1 mM phosphate (control culture) or lacking phosphate (test culture), respectively. The inoculum had an average chromosome copy number of 32.5 copies per cell. In the control cultures the average copy numbers were downregulated to about 10 copies per cell during exponential phase as well as stationary phase (Figure 6A). In the test cultures, the average cell density increased by a factor of 4.1, while the average chromosome copy number dropped by a factor of 4.5 to a copy number of 7.1. Notably, *H. salinarum* also exhibited limited growth in the absence of external phosphate but stopped growing long before the cells became monoploid; in fact, the copy number of 7.1 in starved cells was considerably higher than in the two bacterial species with 4–5 copy numbers. After the addition of 1 mM phosphate, the culture restarted exponential growth, and the chromosome copy number in stationary phase was around seven copies per cell, identical to the copy number after the growth of the control culture. Notably, in this case, the control cultures and the test cultures did not have identical characteristics at the end of the experiment, in contrast to the two bacterial species discussed above. In *H. salinarum*, the test culture had a higher cell density in stationary phase than the control culture (3.7 × 10^9^ versus 1.2 × 10^9^ cells per ml) but had a lower chromosome copy number (about 7 versus about 10 copies per cell).

### 3.5. Characterization of Growth and Genome Copy Numbers of Escherichia coli

It seemed obvious that monoploid species lacking any other phosphate storage polymer (e.g., polyphosphate) cannot exhibit growth in the absence of external phosphate due to the high phosphate content of genomic DNA. Nevertheless, this had not been shown experimentally yet; therefore, we included slowly growing monoploid *E. coli* as a negative control in the present study. Three biological replicates of *E. coli* precultures were grown in synthetic medium with succinate as sole carbon and energy source, which results in doubling times of about two hours and monoploid cells. The cultures were washed, and aliquots were used to inoculate control culture with 6.7 mM phosphate and test cultures lacking phosphate. The inoculum had an origin copy number of 1.4 copies per cell, verifying that the cells were monoploid and a fraction of the population had initiated replication. At the onset of exponential growth of the control culture, the origin copy number slightly increased to two copies per cell, indicating that most or all cells had started replication, while the origin copy number dropped to one copy per cell during stationary phase (Figure 7A). The cell density of the test culture did not increase at all, underscoring that monoploid species cannot grow in the absence of external phosphate (Figure 7B). After the addition of 6.7 mM phosphate, the test culture behaved identical to the control culture, i.e., the origin copy number increased at the onset of growth and dropped to one in stationary phase.

## 4. Discussion

It has been assumed that monoploid species without a phosphate storage polymer like polyphosphate are unable to replicate without an external phosphate and, as a consequence, stop cell division during phosphate starvation. This assumption seems reasonable based on the high phosphate content of genomic DNA, e.g., for the replication of the genome of *E. coli* or *H. volcanii*, about 8 × 10^8^ phosphate molecules are required. However, to our knowledge, this assumption had not been tested previously. Therefore, we decided to characterize the behavior of monoploid *E. coli* cultures in the absence of an external phosphate source. As expected, the cell number remained constant, and thus *E. coli* could serve as a negative control for the other three species. We refrained from characterizing further monoploid species like *Caulobacter crescentus* or *Wolinella succinogenes* during phosphate starvation.

The behavior of three polyploid species during phosphate starvation was characterized in this study. The increase in cell density and the average decrease in chromosome copy number per cell for all four analyzed species are summarized in Figure 8. These results extended our knowledge considerably, because only two species had been characterized previously. Taken together, five polyploid species from five different phylogenetic groups have been characterized until now, i.e., the alphaproteobacterium *Z. mobilis*, the gammaproteobacterium *A. vinelandii*, the cyanobacterium *Synechocystis* PCC 6803, and the two very different halophilic archaea *H. salinarum* and *H. volcanii*, which are from two different genera. The following aspects were of specific interest: (1) do the cells grow under phosphate conditions; (2) if so, do they grow until they become monoploid, or is growth stopped before at a specific ploidy level; (3) are the cells alive after phosphate starvation and do they readily restart growth after phosphate addition; and (4) are genomes degraded during phosphate starvation to liberate phosphate for the synthesis of other phosphate-containing cellular components, e.g., NADP(H), ATP, phospholipids, phophosugars, phosphoproteins. The latter point has been hypothesized to occur during phosphate starvation of *H. volcanii* [52].

It turned out that all five polyploid species (three from this study, two from previous studies) did grow in the absence of an external phosphate source; thus, this behavior seems to be widespread and, until a counterexample is reported, can be assumed typical for polyploid prokaryotes. For *H. volcanii*, it was shown that the cells are able to grow in the absence of external phosphate, but not in the absence of an external source of carbon or nitrogen [52,53]. The fact that for *H. volcanii*, genomic DNA can serve as a storage polymer for phosphate but not for carbon or nitrogen is probably due to the much larger fractions of the latter two elements of other cellular components, i.e., the fraction of cellular dry weight is 3% for phosphor, 9% for nitrogen, and 50% for carbon. It will be interesting to test whether this distinction between phosphor versus carbon/nitrogen is also true for other polyploid prokaryotes.

It should be noted that the precultures in this study were grown in complex medium to guarantee a high chromosome copy number per cell at the start of the growth experiment. All three species also reduced the copy number in synthetic medium in the presence of phosphate (Figure 1A, Figure 3A, Figure 4A and Figure 6A). Therefore, the observed growth in the presence of phosphate was partly based on the usage of the supplied phosphate, and partly due to the reduction in the average chromosome copy number per cell. In contrast, in the absence of phosphate, the growth rested solely on the reduction of the average chromosome copy number per cell, which is only possible for polyploid but not for monoploid species.

Another common feature of the five species was that growth during phosphate starvation stopped before the cells became monoploid and the chromosome copy numbers at this stage varied between 2 (*H. volcanii*) and 7.5 (*A. vinelandii* on sucrose), most often around a copy number of 5. It should be noted that the copy number of *H. volcanii* decreased to two in the reported experiment [52], but growth during starvation often stopped at a copy number of four in later experiments (unpublished data). Growth of *Synechocystis* PCC 6803 during phosphate starvation stopped at a copy number of 4; however, after 10 further days of incubation under phosphate starvation, conditioned cells became monoploid [23]. However, this was not accompanied by further cell divisions, but by the concomitant degradation of other biomolecules, which is known as chlorosis [54,55]. Therefore, this very late additional reduction of the copy number does not indicate growth at the expense of the average copy number per cell, but a cyanobacteria-specific general long-term stress response.

The fact that five out of five characterized polyploid prokaryotic species stopped growth in the absence of phosphate before they became monoploid indicates that all species prefer the genetic advantages of the oligomeric state to further growth. Various different advantages of having several/many chromosomes for prokaryotes have been discussed [15,56]. These include but are not limited to a low mutation rate, a high resistance to double-strand breaks (induced by desiccation or irradiation), gene redundancy, and survival over geological time periods. In this respect, all five species lived a “compromise” between on the one hand making use of the ability to overgrow monoploids that are unable to divide in the absence of external phosphate and on the other hand keeping the genetic advantages of oligoploidy. Characterization of *Deinococcus radiodurans* has revealed that about six genome copies are enough for a very high resistance against irradiation [17,18,19]. It has been shown that *D. radiodurans* can experience a high number of double-strand breaks, which scatter the six chromosomes to hundreds of small fragments, and yet is able to regenerate intact chromosomes very quickly by recombining overlapping fragments [18]. Therefore, we propose that a copy number exceeding, e.g., 10 copies per cell for normal-sized cells does not lead to a further increase in genetic advantages, but that the biological role of the additional copies is to act as “phosphate storage polymers”, which enable the polyploid cells to grow in the absence of external phosphate and thereby gain an advantage not only in comparison to monoploid species, but also to oligoploid species. It should be mentioned that the situation is totally different for giant cells, in which thousands of copies are required to guarantee an even distribution of transcripts and proteins. It should be noted that for normal-sized polyploid species, the amount of phosphate that is sequestered in DNA is higher than the amount of phosphate sequestered in ribosomal RNA, while, in contrast, for normal-sized monoploid species, it is vice versa. For example, *H. volcanii* contains 2.2 × 10^8^ molecules of phosphate in DNA and 1.2 × 10^8^ molecules of phosphate in its ribosomes [52], while fast-growing *E. coli* cells with a generation time of 24 min contain 4 × 10^7^ molecules of phosphate in their DNA and 3.2 × 10^8^ molecules of phosphate in their ribosomes (calculated with 72,000 ribosomes per cell, 4500 nt in the ribosomal RNAs, a copy number of 4.24 of the average genome site between 6.54 origins per cell and 1.94 termini per cell, and a genome size of 4.64 Mbp; data taken from [57]).

The abovementioned aspect would be “phosphate storage” in a broader sense, because genomic DNA is not degraded but is passed on to the daughter cells. To be a “phosphate storage polymer” also in a second, strict sense, genomic DNA would have to be degraded under phosphate starvation conditions to liberate phosphate for other biomolecules, e.g., ATP, NADP(H), phospholipids, phosphosugars, phosphoproteins. For *H. volcanii*, it has been hypothesized that genomic DNA fulfills both roles under phosphate starvation conditions because more than 30% of the genome-bound phosphate was missing after the growth under phosphate starvation conditions [52]. To obtain insight on whether or not this might also be possible for the species characterized in the current study, in Appendix A, phosphate balances for cells prior to and after growth in the absence of phosphate are summarized. For all three species analyzed in this study, the fraction of missing phosphate molecules was much smaller than for *H. volcanii*: it ranged from 2.7% to 10.4%. Therefore, only a small fraction of genomes was degraded to liberate phosphate, or genome degradation did not occur at all. It is hard to envision that the intracellular concentrations of central phosphate-containing molecules like ATP or NADP were reduced by factors of three to seven; however, it cannot totally be ruled out. No changes in cell volume was observed during cell counting, indicating that the cells did not keep up the concentrations via reducing the cell volume. Putative alternatives include the degradation of a fraction of the ribosomes or liberation of phosphate from other biomolecules like phospholipids. Future studies are needed to gain a more complete overview of the phosphate balance for these three species during growth in the absence of external phosphate.

Taken together, the results concerning the first two aspects under investigation were uniform for all five species; namely, all species grew in the absence of external phosphate and all species stopped growth before they became monoploid. In addition, all four tested species were alive after phosphate starvation and readily started exponential growth upon the addition of phosphate. This point is unclear for *Synchocystis* PCC6803, because phosphate addition to starved cells was not included in the starvation experiment [23]. However, cyanobacteria are known to withstand various stresses [58,59,60], and thus it can be expected that they would have had the ability to restart growth after phosphate addition. Concerning the fourth and last aspect, results were not uniform for the different species, but differed considerably. For three species, i.e., the two haloarchaea *H. volcanii* and *H. salinarum* and for *A. vinelandii* grown on sucrose, the results are at least compatible with the idea that a minor fraction of the genomes were degraded during growth in the absence of phosphate to mobilize phosphate. In these cases, it seems worthwhile to use alternative experimental approaches to clarify the question in the future. One possibility could be to visualize one genomic site fluorescently, which would enable direct counting of chromosomes. This technique has been established for *Bacillus subtilis*, *Caulobacter crescentus*, *Vibrio cholerae*, and several other species [61,62,63,64]. However, it is not available for the three respective species, and, in addition, until now, it has never been applied to polyploid species that would require an extremely high resolution to detect 20–30 fluorescent foci in a small prokaryotic cell simultaneously.

A side aim of this project was to clarify the ploidy level of *A. vinelandii*, for which very contradicting results have been reported. Using different experimental approaches, it was found to either upregulate the chromosome copy number during the growth curve and become highly polyploid with 80 copies of the chromosome in stationary phase [24,39], or to be monoploid [39]. The latter opinion was obtained with genetic approaches, e.g., the generation of homozygous mutants or the way mutations segregated were taken as genetic proof for monoploidy [37,39]. However, since then, it has become clear that at least two polyploid prokaryotes, the methanogenic archaeon *Methanococcus maripaludis* and the haloarchaeon *H. volcani*, exhibit highly efficient gene conversion [8,65,66]. Gene conversion equalizes homologous, but nonidentical gene copies [67]. If it is selected, one initial mutant copy can overwrite several to many wild-type copies; thereby, independent segregation of copies does not occur, and mutant copies become phenotypically visible despite the initial much higher copy number of the wild-type copy. Therefore, if highly efficient gene conversion would also operate in *A. vinelandii*, the observed segregation patterns of mutant copies would not be indicative for monoploidy.

In complex medium, 20.6 genome copies in exponential phase and 19.6 genome copies in stationary phase were observed (see above). Therefore, we also found that *A. vinelandii* is polyploid when grown in complex medium, in congruence with earlier reports [24]. However, both the absolute copy numbers as well as their dynamic regulation found in this study differed from previous reports. We found neither the low copy number of only four in exponential phase nor the high increase to 80 copies per cell in stationary phase. The differences might well be due to variations in the medium composition, aeration, other physical or chemical parameters, or strains.

The genome copy number of *A. vinelandii* is lower in synthetic medium than in complex medium, in agreement with an earlier publication [24]. However, a high increase to 40 genome copies was observed at the onset of exponential growth, which has not been described before for *A. vinelandii*. Notably, a high increase in copy number at the end of the lag phase and onset of growth with a decrease during exponential phase has recently been described for *V. natriegens* [47]. In addition, an increase in genome copy number prior to the onset of growth has also been described for the cyanobacterium Synechococcus elongatus PCC 7942 [5]. It will be interesting to reveal whether this over-replication of chromosomes before the onset of cell division is widespread also in other polyploid prokaryotes.

## 5. Conclusions

In this study, three additional polyploid prokaryotic species were incubated in the absence of an external phosphate source. Because two species had been characterized in previous studies [23,52], this increased the number of characterized species to five, which belong to three bacterial and two archaeal genera. The following behavior was observed for all five species and can thus be predicted to be typical for and widespread among polyploid prokaryotes:(1)All five species exhibited limited growth during phosphate starvation. The cell densities increased by 2.9-fold to 6.7-fold, with a concomitant decrease in the chromosome copy number per cell.(2)All five species stopped growth before they became monoploid with a chromosome copy number of around five.(3)Out of the five species, all four tested species readily restarted exponential growth when phosphate was added to the starved cultures and thus survived the starvation.

## Figures and Tables

**Figure 1 microorganisms-11-02267-f001:**
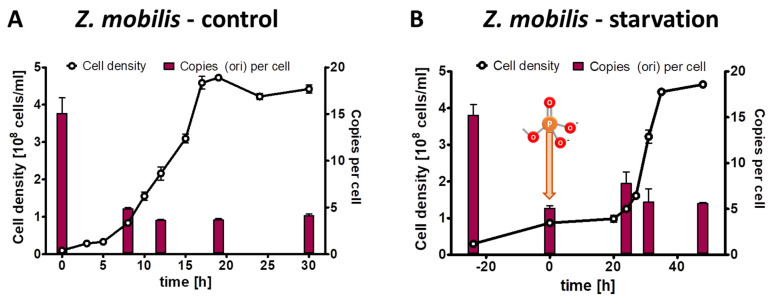
*Z. mobilis* cultures were grown in the presence of 14 mM phosphate (**A**) or incubated in the absence of external phosphate (**B**). Phosphate was added to the starved culture at the time indicated in (**B**). Aliquots were removed at the indicated time points, the cell densities were quantified using a Neubauer counting chamber, and the number of chromosomes were quantified using qPCR. Three biological replicates were performed, and average results and their standard deviations are shown.

**Figure 2 microorganisms-11-02267-f002:**
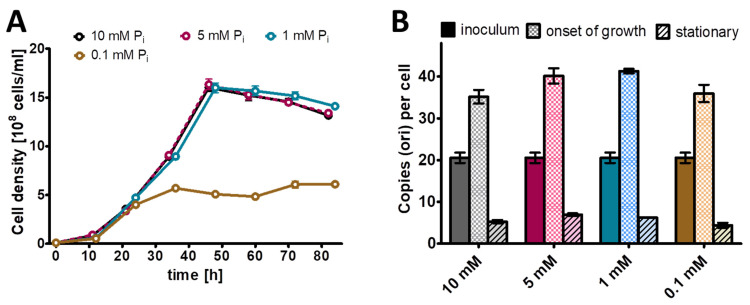
*A vinelandii* cultures were grown in the presence of the indicated four concentrations of phosphate. (**A**) Growth curves. (**B**) Chromosome copy numbers determined at three time points, as indicated. Three biological replicates were performed, and average results and their standard deviations are shown.

**Figure 3 microorganisms-11-02267-f003:**
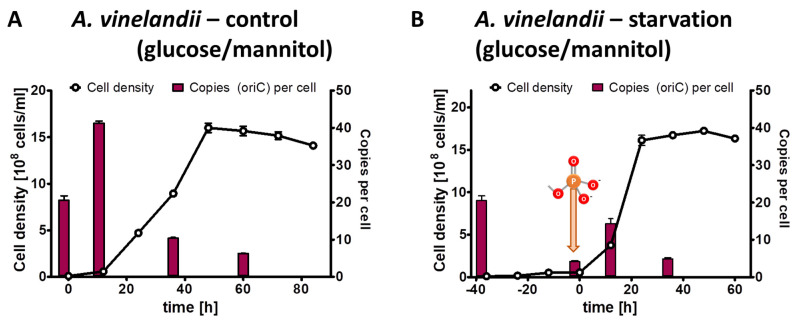
*A. vinelandii* cultures were grown in Burk’s synthetic medium with glucose/mannitol the presence of 1 mM phosphate (**A**) or incubated in the absence of external phosphate (**B**). Phosphate was added to the starved culture at the time indicated in (**B**). Aliquots were removed at the indicated time points, the cell densities were quantified using a Neubauer counting chamber, and the number of chromosomes were quantified using qPCR. Three biological replicates were performed, and average results and their standard deviations are shown.

**Figure 4 microorganisms-11-02267-f004:**
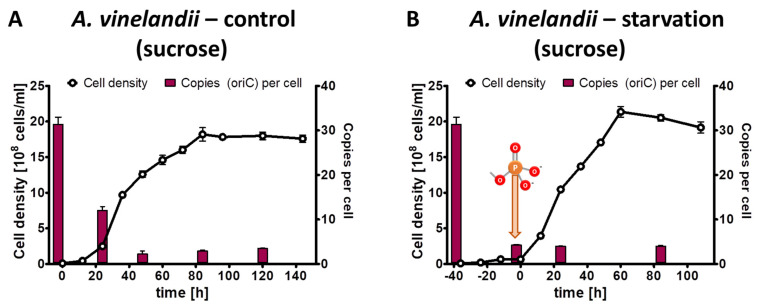
*A. vinelandii* cultures were grown in Burk’s synthetic medium with sucrose in the presence of 1 mM phosphate (**A**) or incubated in the absence of external phosphate (**B**). Phosphate was added to the starved culture at the time indicated in B. Aliquots were removed at the indicated time points, the cell densities were quantified using a Neubauer counting chamber, and the number of chromosomes were quantified using qPCR. Three biological replicates were performed, and average results and their standard deviations are shown.

**Figure 5 microorganisms-11-02267-f005:**
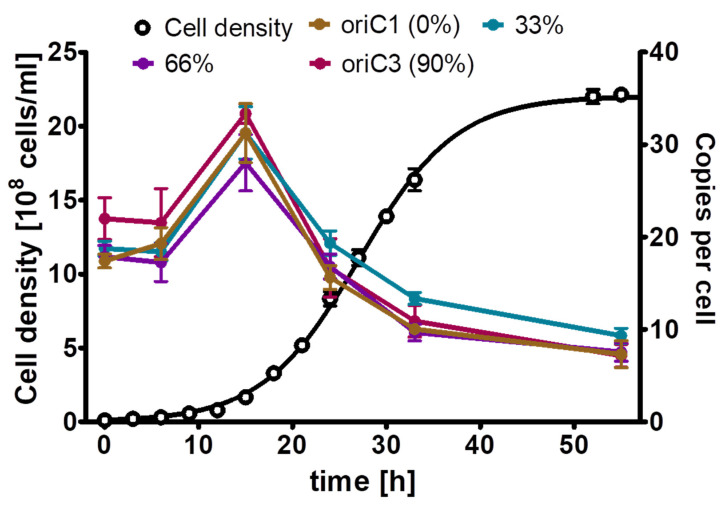
*H. salinarum* cultures were grown in complex medium. At the indicated time points, aliquots were removed, the cell densities were determined using a Neubauer counting chamber, and the copy numbers of the four indicated sites of the major chromosome were quantified using the qPCR method. Three biological replicates were performed, and average results and their standard deviations are shown.

**Figure 6 microorganisms-11-02267-f006:**
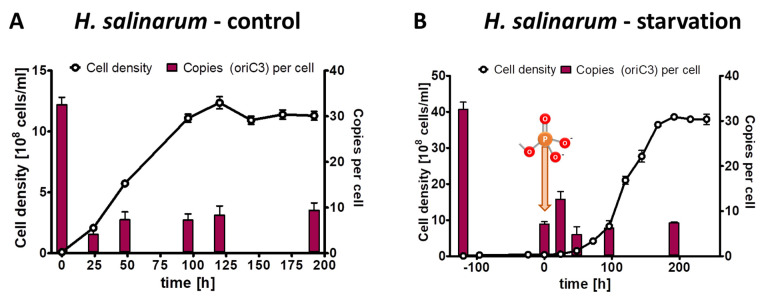
*H. salinarum* cultures were grown in the presence of 1 mM phosphate (**A**) or incubated in the absence of external phosphate (**B**). Phosphate was added to the starved culture at the time indicated in (**B**). Aliquots were removed at the indicated time points, cell densities were quantified using a Neubauer counting chamber, and the number of chromosomes were quantified using qPCR. Three biological replicates were performed, and average results and their standard deviations are shown.

**Figure 7 microorganisms-11-02267-f007:**
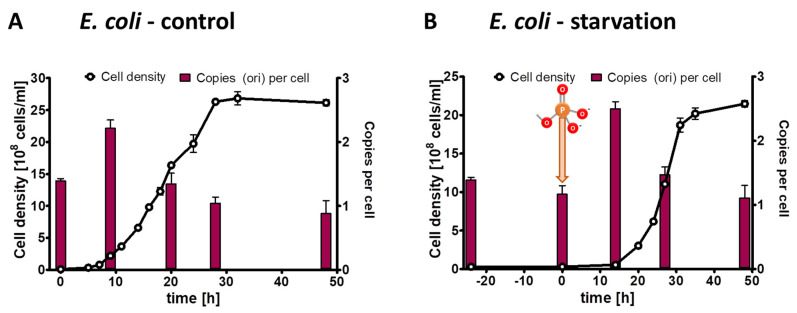
*E. coli* cultures were grown in the presence of 6.7 mM phosphate (**A**) or incubated in the absence of external phosphate (**B**). Phosphate was added to the starved culture at the time indicated in (**B**). Aliquots were removed at the indicated time points, the cell densities were quantified using a Neubauer counting chamber, and the number of chromosomes were quantified using qPCR. Three biological replicates were performed, and average results and their standard deviations are shown.

**Figure 8 microorganisms-11-02267-f008:**
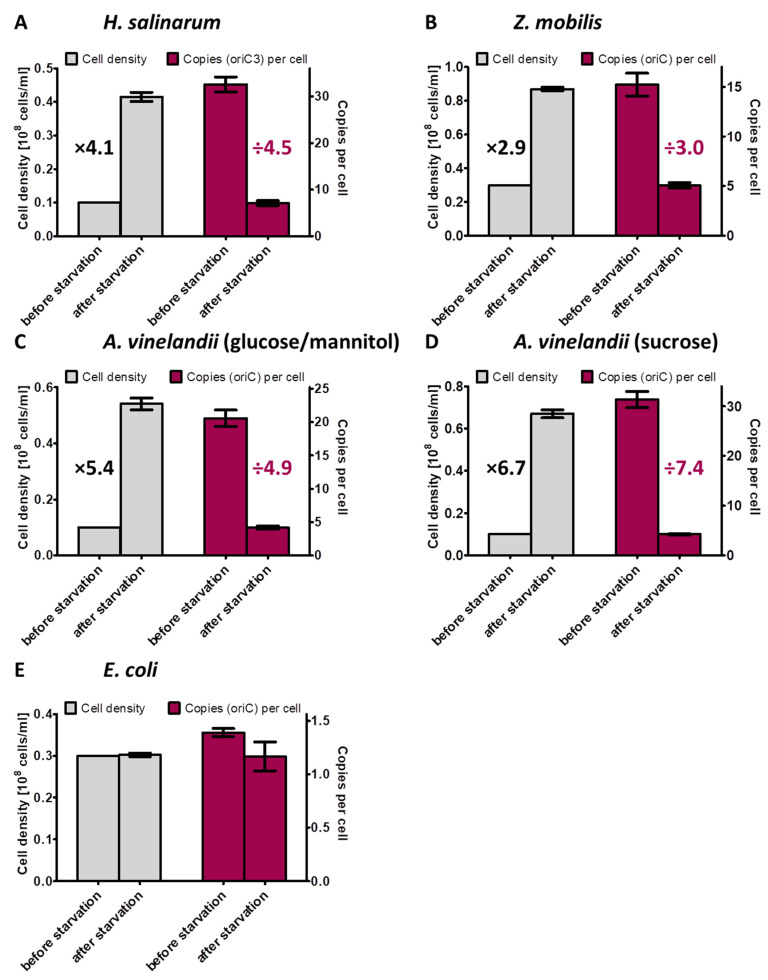
Overview of the increase in cell densities and the concomitant decrease in chromosome copy numbers per cell during the incubation in the absence of an external phosphate source for the four prokaryotic species characterized in this study.

## Data Availability

Not applicable.

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
