# Peer review of "One Advantage of Being Polyploid: Prokaryotes of Various Phylogenetic Groups Can Grow in the Absence of an Environmental Phosphate Source at the Expense of Their High Genome Copy Numbers"

_microorganisms, 2023, doi:10.3390/microorganisms11092267_

Round 1

Reviewer 1 Report

The study by Brück and colleagues addresses the possible use of chromosomal DNA in polyploid bacteria as a source of phosphorus (P) in case of P limitation.

The authors tested three polyploid bacteria to supplement the previous data on two other strains. Additionally, the authors tested contradicting results regarding the polyploidy of one of their strains (A. vinelandii). 

The questions addressed here are of high significance to get a more complete understanding of the multiple roles and benefits of polyploidy in Bacteria and Archaea. 

I have made my remarks on the attached annotated PDF file of the paper, hence I will put here only several more major comments.

1) In all tested strains presented in this paper, a drop in the number of chromosomes per cell is observed both in the control group as well as in the P-starved group. Yet, the authors do not address the drop in the control group. Since both experimental groups behave the same in this respect it is hard to attribute the change in chromosomes per cell in the P-starved group to degradation of DNA to be used as a source of P.

This may be a result of changing from a complex media in the pre-culture to a synthetic media in the experiments. I am not sure how to address this without new experiments. If conducting new experiments is not feasible, the authors should address this at least in writing in the text.

2) With regard to Table 1 and the P budget. It's a very confusing section that considers only part of the P storage in cells. The authors also conclude that not much can be concluded from the experiments of how much DNA-derived-P was used for cell growth. Therefore, I suggest removing this table and this section from the discussion.

3) The results section contains in some cases repetition of the methods section as well as some sentences of discussion. With respect to methods. These should be removed from the result. As for the discussion, either these are removed (this will not affect the quality or clarity of the paper in my opinion) or the results and discussion section should be merged, which is much more work.

Overall, I am looking forward to see this work published and I trust it will make a significant addition to the study field of polyploid bacteria. 

Author Response

I have uploaded a file with the answers to your comments, because I could use color to discriminate comments and answers.

Reviewer 2 Report

Genomic DNA has a high phosphate content, monoploid prokaryotes need an external phosphate source or an internal phosphate storage polymer for replication and cell division. This work to test this view is widespread in and typical for polyploid prokaryotes, the other three species fo testing. The results support the view, and is interesting. The review is enjoying. There are some comments as below:

1. In introduction, how to effect the gene copies by proposed, this is the important point for this work. I suggest author to add the part and detailed description. add the number of chromosome.

2. in method, if author add the following of the step, maybe will introduce the method.

3. the results is better. and the figure is clearly.

4. in discussion, the part is better.

the english is better

Author Response

Thank you for this very positive review and for finding reading of the manuscript enjoying. Following your suggestion, I have added information about the gene copy numbers to the Introduction. In the Method section I have added a general introduction to the qPCR method for the quantification of the number of chromosomes per cell.

Reviewer 3 Report

The study presented by Patrick Brück et al. investigates whether or not three polyploid bacteria from different phylogenetic groups can grow in the absence of phosphate in the environment. This is based on the assumption that bacteria with a high number of genome copies can degrade their genome to obtain phosphate. Indeed, the results obtained by the researchers suggest that the three species investigated in the study can grow in the absence of environmental phosphate by reducing genome copy number. However, the methodology used does not guarantee the reliability of the results obtained.

While the concept itself is interesting, the study raises important questions. The first and most important is the methodology used. In order to count the number of genomes, the authors of the study do not explain the criteria they used to select the genomic region for amplification in the three bacterial species. For this analysis, it is necessary to select genes that are present in only one copy in these genomes, otherwise, the number of genome copies per cell may be either overestimated or underestimated. Another problematic aspect of the following protocol is the use of two PCR steps. One of them is used to amplify a 1Kb region, which is then analyzed by qPCR to estimate the genome copy number. This preliminary PCR can introduce a great deal of variability into the downstream analysis. Furthermore, it is not explained exactly how this PCR is performed from genomic DNA, was the genomic DNA purified with a purification kit?  This preceding PCR may represent a significant bias in downstream analyses. PCRs from genomic DNA, if not properly purified, may contains ions and other cytosolic substances from bacterial cells that affect PCR performance. For example, in the cyanobacterium Synechocystis, PCR efficiency is compromised if genomic DNA is not thoroughly purified due to the high Mg content in chlorophyll. The same can happen when genomic DNA is recovered from these three bacteria, introducing other interfering elements that affect PCR performance. In addition, the text lacks a clear explanation of how PCR amplification of the 1-kb fragment from genomic DNA obtained from cytoplasmic extract is normalized based on cell number. The method requires a more complete description, including additional details.

Another problem with the experimental design is that researchers do not use the same time points for obtaining DNA samples for PCR analyses in the control culture and the starvation culture, nor do they use the same experimental time periods. For example, Figure 1 shows that the cells in the control culture grew for 30 hours in phosphate-containing medium, whereas they grew for only 20 hours in the starvation culture. Also, during growth in the presence of phosphate, four analysis points were taken at 8, 12, 18, and 30 hours, while under phosphate starvation only a single point was taken at 20 hours. The comparisons should be made using similar points in time. I also do not understand why the initial optical densities are not similar in both conditions. For example, in Figure 1A, the initial OD is much lower than in panel B. If the difficulty in obtaining intermediate analysis points under phosphate starvation is due to the lack of cells for sufficient extraction of genetic material, I suggest starting the growth curves with higher initial optical densities to ensure sufficient DNA extraction during phosphate starvation. For example, inoculate the cultures at an optical density of 1. Afterward, proceed with the experiment under both phosphate-deficient (-P) and phosphate-rich (+P) conditions for extended periods of 30, 50, or 60 hours. This would provide a more comprehensive understanding of the changes in genome copy numbers and allow for a meaningful comparison between the two conditions. 

Another significant concern that the researchers should take into account in the current study is the presence of available phosphate in the phospholipids of cell membranes. Could these bacteria be degrading parts of cell membranes to get phospholipids instead of degrading genetic material? To answer this question, I suggest not only counting the number of cells when taking cell samples, but also measuring the diameter of the cells. This allows monitoring of cell size and consequently assessment of reduction in cellular phospholipid content. Such a reduction may indicate phospholipid recycling for intracellular phosphorus recovery.

Line 336 “downregulated” change by decreased. 

Finally, I think it's confusing that the conclusions of the study include results of analyses published in other studies. The current wording may give the impression that this study examined a total of five polyploid species by including previous research with two other polyploid species. I suggest that the conclusions of the study be limited exclusively to the three species analysed in this study.

Taking all these considerations into account, I believe that the manuscript needs to address the experimental issues raised before being considered for publication.

Author Response

I have uploaded a file with the answers, because I could use color-codong for an differentiation between comments and answers.

Round 2

Reviewer 3 Report

I thank the authors for their quick and detailed response to each point. I believe my concerns have been thoroughly addressed. 

In my opinion, the article is now ready for publication